# The Pivotal Role of Quantum Dots-Based Biomarkers Integrated with Ultra-Sensitive Probes for Multiplex Detection of Human Viral Infections

**DOI:** 10.3390/ph15070880

**Published:** 2022-07-17

**Authors:** Seyyed Mojtaba Mousavi, Seyyed Alireza Hashemi, Masoomeh Yari Kalashgrani, Navid Omidifar, Chin Wei Lai, Neralla Vijayakameswara Rao, Ahmad Gholami, Wei-Hung Chiang

**Affiliations:** 1Department of Chemical Engineering, National Taiwan University of Science and Technology, Taipei City 106335, Taiwan; kempo.smm@gmail.com (S.M.M.); vijayrao@mail.ntust.edu.tw (N.V.R.); 2Nanomaterials and Polymer Nanocomposites Laboratory, School of Engineering, University of British Columbia, Kelowna, BC V1V 1V7, Canada; s.a.hashemi0@gmail.com; 3Biotechnology Research Center, Shiraz University of Medical Sciences, Shiraz 71468-64685, Iran; masoomeh.yari.72@gmail.com; 4Department of Pathology, Shiraz University of Medical Sciences, Shiraz 71468-64685, Iran; omidifar@gmail.com; 5Nanotechnology and Catalysis Research Centre (NANOCAT), Level 3, Block A, Institute for Advanced Studies (IAS), Universiti Malaya (UM), Kuala Lumpur 50603, Malaysia; cwlai@um.edu.my

**Keywords:** viral infections, quantum dots, ultra-sensitive probes, biomarkers, multiplex detection

## Abstract

The spread of viral diseases has caused global concern in recent years. Detecting viral infections has become challenging in medical research due to their high infectivity and mutation. A rapid and accurate detection method in biomedical and healthcare segments is essential for the effective treatment of pathogenic viruses and early detection of these viruses. Biosensors are used worldwide to detect viral infections associated with the molecular detection of biomarkers. Thus, detecting viruses based on quantum dots biomarkers is inexpensive and has great potential. To detect the ultrasensitive biomarkers of viral infections, QDs appear to be a promising option as biological probes, while physiological components have been used directly to detect multiple biomarkers simultaneously. The simultaneous measurement of numerous clinical parameters of the same sample volume is possible through multiplex detection of human viral infections, which reduces the time and cost required to record any data point. The purpose of this paper is to review recent studies on the effectiveness of the quantum dot as a detection tool for human pandemic viruses. In this review study, different types of quantum dots and their valuable properties in the structure of biomarkers were investigated. Finally, a vision for recent advances in quantum dot-based biomarkers was presented, whereby they can be integrated into super-sensitive probes for the multiplex detection of human viral infections.

## 1. Introduction

Nanoscale analytical tools are having a remarkable impact on transforming current analytical methods into diagnostic approaches by transforming their sensing module for the detection of biological molecules such as viruses (cold, flu, chickenpox, HIV, and SARS-CoV-2). For viral particles, contemporary biosensing platforms must be continuously upgraded due to technological limitations and biological barriers. Accurate and rapid detection of viruses is critical to prevent the spread of pathogens and ensure rational and effective treatment management in the early stages of infection. Using nanobiosensors for single-shot analysis of biological samples is also a standard method for detecting viruses [1,2].

Nanobiosensors are devices for measuring a biochemical or biological reaction using electronic, optical, magnetic, or data-gathering technology through a probe, and they possess several advantages, including fast response, high accuracy, achievable process, miniaturization, and high sensitivity, thus making detection very effective [3,4]. Hence, there are fewer detection limitations with nanotubes, nanowires, nanomembranes, or nanotextured surfaces [5,6]. The importance of these devices is evident in diagnosing diseases and human health. These devices can instantly describe the physiological state of the disease [7]. The use of nanomaterials in biomarkers has been considered due to their unique surface, mechanical, optical, electrical conductivity (EC), biological and magnetic properties, and catalytic activity [5,8]. Biomarkers are important tools for diagnosing, predicting, and monitoring disease progression and drug development in nanobiosensor devices [9]. The classification of nanosensors for virus detection according to measurable characteristics and detection mode results in different categories such as biological, electrochemical, piezoelectric, magnetic, optical, and thermal detection. Table 1 provides an overview of electrochemical and optical nanobiosensors in viral infections.

A biomarker in the medical field is a measurable indicator for diagnosing the presence of a disease, which sometimes also indicates the severity of the disease [10]. Biomedical markers refer to anything which can be used as an indicator of a specific disease or some physical condition of a living organism. A biomarker can also be a substance that traces the symptoms of a particular disease. For example, antibodies are the signs of an infection and are detected by a biomarker. Biomarkers indicate the expression changes or protein status associated with a disease or its progression. In other words, biomarkers can be biological properties or molecules identified by testing body parts, such as specific tissues or blood tests, showing disease or normal processes in the body [11,12]. These markers can also be particular cells, genes, gene products, enzymes, or hormones. Sometimes the functions of some organs or certain changes in biological structures can be used as biomarkers.

Although the term biomarker is relatively new, it has been used extensively in clinical research and detection [13,14]. In addition to expanding methods for detecting infectious diseases, there is an increasing need to find biomarkers based on QDs with higher sensitivity and the ability to determine the extent and course of disease activity. The features of quantum dot-based biomarkers include being a non-invasive, sensitive, and cost-effective method, high stability in the sample, high specificity for detection of disease, identification before symptoms of viral diseases appear, as well as the relevance in citing and translating from animal models to humans [15]. The first glimpse of the vast potential of quantum dot-based biomarkers as probes for studying biological systems suggests that quantum dot-based biomarkers can be dissolved in water and conjugated with biological molecules. QD-based biomarkers have advantages over organic dyes and fluorescent proteins, such as multicolor fluorescence emission, improved brightness, and resistance to light bleaching. These advantages could improve biological detection and imaging sensitivity by at least one to two orders of magnitude [16].

**Table 1 pharmaceuticals-15-00880-t001:** Overview of electrochemical and optical nanobiosensors in viral detection.

	Detection Method	Nanomaterial	Limit of Detection	Type of Virus	Ref.
Electrochemical nanobiosensors	SWCNTs	Nanotubes	102 CFU/mL	Bacillus subtilis	[17]
	Change in output voltage	-	6.9 copies/µL of viral RNA	*SARS-CoV-2*	[18]
	Amperometric readings	-	-	*SARS-CoV-2*	[19]
	FET sensor, transfer curve shift	-	2.29 fM–3.99 fM	SARS-CoV-2 RNA	[20]
	Amperometry	Silver graphene QDs (Ag/GQDs)	1ZM	Legionella	[21]
	glip-T	1,6-Hexanedithiol and chitosan stabilized gold nanoparticle	0.32 ± 0.01 × 10-[14]	Invasive Aspergillosis (IA)	[22]
Optical nanobiosensors	Fluorescence	CdTe QDs	0.13 µg mL^−1^	Citrus tristeza virus (CTV)	[23]
	LSPR, plasmonic photothermal heating (dual sensor)	-	0.22 ± 0.08 pM (2.26 × 104 copies of viral RNA)	Coronavirus 2	[24]
	Fluorescence	-	12.6 nM of spike RBD	COVID-19	[25]
	Terahertz plasmonic sensor	-	4.2 fM	*SARS-CoV-2*	[26]
	SPRi	AuNPs induced with QDs	0.03 pg/mL and 0.4 pg/mL, 10 PFU/mL	Influenza	[27]
	Fluorescence	Nanobeads	102–103 CFU/mL	*E. coli*	[28]

This review study aims to explore advances in quantum dot-based biomarkers integrated with ultra-sensitive probes for multiplex detection of human viral infections and highlight future work in this area. Additionally, the classification of QD-based nanobiosensors based on their detection technique, structure, and function for detecting different viruses was investigated. Figure 1 shows the QD-based nanosensors for virus detection. In addition, the QD-based biomarkers, viral infections detected by QD-biomarkers, mechanism of QD-based biomarkers, and multiplex detection of viral infections were evaluated.

## 2. The Chemistry of Semiconductor QDs, Carbon QDs, and Graphene QDs and Their Functionalization Strategies

A quantum dot is a region in a semiconductor crystal that includes electrons, holes, or both in three dimensions. All three dimensions of matter are on the nanometer scale. The main feature of these dots is light scattering. Their dimensions are so small that the laws of classical physics cannot explain the properties of matter; only quantum physics can explain the behavior of matter. Quantum dots are a unique class of semiconductors because of their small size. The importance of quantum dots as semiconductor gates is that the electrical conductivity of these materials can be changed by external stimuli, such as electric fields or light radiation, to the extent that they switch from being non-conducting to conducting and acting as switches. Compared to real atoms, semiconductor quantum dots have the unique feature that the number of free electrons can be changed using external instruments [29,30]. This simple method creates artificial atoms with 3, 2, 1, or more electrons. Therefore, adding or subtracting electrons to quantum particles will produce a wide range of synthetic materials. However, it must be remembered that the synthetic materials produced by this method will not have all the properties of the original materials. The materials likely produced will ultimately consist of nanorobots under human command in appearance and function. Therefore, the two properties of quantum confinement effects and edge effects are the fundamental properties of graphene and carbon quantum dots, which have unique chemical and physical properties such as non-toxicity, biocompatibility, and stable optical properties. These fundamental properties result in many applications in various fields such as detectors, medicine, optoelectric devices, batteries, supercapacitors, catalysts, and sensors [31]. Research on graphene and carbon quantum dots is still in its infancy, and many challenges remain to be solved. Geim and Ponomarenko first fabricated graphene quantum dots in 2008, which are two-dimensional structures with a graphene lattice less than ten layers thick and less than a few tens of nanometers in size. These structures have many novel properties, including a unique luminosity due to the effects of quantum confinement [32]. Carbon quantum dots were first produced in 2004 to purify single-walled carbon nanotubes. Carbon quantum dots have a spherical morphology of less than 20 nm. Later, in 2006, the above structure was also successfully fabricated by laser ablation of graphite powder, thus taking a new step towards identifying and using more of these materials [33]. Pure graphene and carbon quantum dots have many limitations that limit their applications. To extend their applications to different fields, these structures can be functionalized by various methods, such as doping with different atoms, forming composites with minerals or polymers, controlling the size and changing the morphology, and optimizing them for specific applications by changing their chemical, optical, and electronic properties. Doping semiconductor materials is an essential process in the semiconductor industry because it can change the materials’ key physical, chemical, and electronic properties and improve their performance in various applications. Graphene and carbon quantum dots are no exception to this rule, as their multiple properties can be improved in this way. So far, most phosphorus, nitrogen, sulfur, selenium, chlorine, fluorine, and boron atoms have been used to dope the aforementioned quantum dots, and the results have been studied using various approaches, such as increasing the band gap and shifting the optical absorption peak, as well as changing the intensity of the photoluminescence emission. Another strategy is to change the optical, physical, and chemical properties of graphene and carbon quantum dots by controlling the size and shape of these particles. In addition, there are various methods in which the fabrication conditions and pH factors are changed to alter the size and morphology of graphene quantum dots. These methods have shifted the band gap from the blue region of the spectrum to the red region [34].

### The Comparative Merits and Demerits of Semiconductor QDs, Carbon QDs, and Graphene QDs

For quantum dots made of semiconductor materials, the concepts related to capacitance, conductivity, and the forbidden band for semiconductor materials also apply to quantum dots. Nevertheless, a critical difference between them is that electron transfer occurs easily in bulk materials because there is enough space for the electrons to move. In quantum dots, however, this electron transfer occurs at a minimal radial distance called the “Bohr radius”. When the dimensions of the quantum dots or quantum crystals are as small as the Bohr radius, the electron can no longer move as easily in the matter, and the laws of electron movement and transfer change dramatically. This leads to unique optical properties, including the effect of light absorption and reflection in semiconductor crystals with dimensions in the range of the Bohr radius. Another feature of semiconductor quantum dots is that a change in the number of atoms causes a change in the forbidden band; in other words, it changes the energy difference between the conducting layer and the capacitance. In addition to the number of atoms, the way they are arranged at the quantum dot level affects the magnitude of the energy difference [35]. Carbon quantum dots (CQDs) and graphene quantum dots are a new class of carbon nanomaterials that have recently emerged and are considered potential competitors of semiconductor quantum dots due to their low toxicity, environmental friendliness, and low fabrication cost. Although many essential and practical advantages have been identified for the above structures, further research is still being conducted to improve the recognition of these advantages and their use in various applications. Mass graphene and carbon quantum dots production at a relatively low cost is required to meet industrial needs. Of course, there are challenges to be overcome in their industrialization and mass production. The reported quantum efficiency of quantum dots is lower than that of conventional semiconductor quantum dots. Therefore, the low quantum efficiency is another challenge that requires further research. Another problem is the accuracy of the results of optical studies, which have always been controversial, and only changing the fabrication method changes the comparison of the results. This problem has led to a lack of understanding of the mechanism of photoluminescence and the associated analysis of graphene and carbon quantum dots [36,37]. The following Table 2 shows the advantages and disadvantages of semiconductor, carbon, and graphene quantum dots.

## 3. QD-Based Nanobiosensors

In recent years, various nanomaterials have been used in biosensors to increase selectivity and accuracy. QDs, widely used in bioimaging and bioassay, are known as zero-dimensional nanomaterials. They are also known as semiconductor nanocrystals [46,47]. Size-dependent emission due to quantum confinement and attractive luminescence due to their small size and quantum effect are among the features of QDs. The binding of QDs to specific biomolecules, such as oligonucleotides, proteins (peptides, antibodies, or enzymes), and polymers, is possible by having these features together [48]. Chen et al. examined the rapid and sensitive detection of avian influenza (AIV) virus in the H5N1 subtype by developing QDs-based fluoroimmunoassay assays [49,50]. In the virus detection system, QDs act as energy donors; thus, the design and synthesis of QD-based nanobiosensors must consider optical efficiency, photostability, fluorescence background, and energy transfer efficiency [51]. The following sections discuss some of the essential types of quantum dot-based nanobiosensors. Figure 2 represents the classification of nanobiosensors for viral detection.

### 3.1. Fluorescence

An optical phenomenon in which photon absorption at a shorter wavelength causes propagation at longer wavelengths is called fluorescence [52]. Using fluorescent nanomaterials as probes for bioanalytical applications is a new and emerging technique due to their high stability, sensitivity, large band gap, superior water solubility, and excellent fluorescence properties [53]. Fluorescent biosensors can detect viruses due to various parameters such as intensity, energy transfer, lifetime, and quantum efficiency [54]. In fluorescence-based light sensors, the inorganic semiconductors QD, CD, organic conjugated polymer nanoparticles, and graphene nanostructures are several different light-emitting materials (fluorophores) [55]. The fluorescence sensors have become the most common optical technique due to their low detection limitations and format simplicity. Since the quenching phenomenon occurs in the presence of sample components and some materials (analytes), QD-based probes have low selectivity, based on the existing reports [56]. Fluorescence-based optical sensors have excellent performance in detecting viruses due to the availability of commercial fluorescent probes and advanced optical elements. The change in the fluorescent signal of the probe is measured to detect the virus.

The change in the fluorescent signal of the probe is measured to detect the virus. However, photo quenching and photobleaching usually occur with fluorescent molecules under high-power laser irradiation or long-term illumination. Therefore, an ultrasensitive fluorescent sensor must be constructed using suitable fluorescent molecules with adjustable laser power [57]. Compared to commercial fluorescent molecules, colloidal semiconductor nanomaterials called QDs have optical properties such as adjustable emission wavelength, high quantum efficiency, and excellent anti-photovoltaic ability [55,58]. Therefore, nanoparticle doping with semiconductor QDs or lanthanide chelates is involved in designing fluorescence-based assays [59]. Devices to detect pathogenic organisms using natural biocompatible substances with a proper substrate are called nanomaterials-based biosensors. Usually, a suitable data processing system is connected to these platforms [60]. One of the reasons that lead to the bioconjugation of QDs to the biosensor structure is a growing interest in detecting viruses by utilizing nanomaterials-based biosensors. Low cytotoxicity, superior biocompatibility, high photostability, and accessible surface modification are among the amazing properties of fluorescent QDs [61,62,63].

Fluorescent biosensors based on QDs are noted for their ability to measure different analytes simultaneously, increased stability, and high sensitivity. The fluorescence-based resonant energy transfer is the most widely used method for QD-based biosensors, as shown in Figure 3. For example, depending on the effects of quantum confinement, adjustment from the UV region to the near-infrared region (NIR) can be made for the emission wavelength used for QDs in fluorescence sensors [64]. Pan et al. used NIR emitting QDs to modify avian influenza (H5N1p) pseudo-typed virus through biorthogonal chemistry [65]. Ngeontae et al. fabricated a QD-based biosensor to detect abnormal adenosine-5′-triphosphate (ATP) levels using fluorescent techniques [66].

### 3.2. Nanowire

When forming heterostructures in small dimensions, nanowires (NWs), including QD, offer several degrees of freedom thanks to their morphology and size [67]. Nanowires are the best option for creating robust, sensitive, and selective electrical detectors for bio-bonding purposes. Through susceptible and direct electrical methods, devices containing nanowires are best used for optical excitation, electron transfer, and as carriers as a means of determining the biological and chemical species [68]. To study the environment and in-body sensing, the main factor in developing remote nanobiosensors is the delicate nature of nanowires [69]. Nanobiosensing devices made of nanowires or fibers at the nanoscopic scale are called nanowire-based sensors, which are covered by bionanowires (macromolecules), including proteins, fibrin, and DNA molecules. The current flows in nanowires are very close to the surface, where the current flow in any one-dimensional system is susceptible to minor disturbances [70,71]. The diameter of these nanowires is the size of biological macromolecules. Unlabeled direct electrical reading is achieved through the ability to attach analytes to the surface and combine the tunable conducting properties of semiconductor nanowires [72]. These sensors operate on the principle of ion-selective field-effect transistors and rely on the interaction of external charges with carriers in the nearby semiconductor, which increases sensitivity to low ionic strength. QD nanowires formed via genetically engineered non-toxic virus will be used as energy channels. These QD-coated viral nanowires will be biologically conjugated to a light-harvesting antenna (up-conversion nanoparticles) at one end and QD receivers at the other, forming biologically-templated energy circuits as depicted in Figure 4 [73].

### 3.3. Graphene

Graphene has attracted much attention experimentally and theoretically from the scientific communities in recent years due to its unique physical properties. In nanobiosensors, graphene is produced through the GO reduction method [5,74]. The graphene and GO samples obtained so far are generally micron-sized or larger. Finally, little is known experimentally about the properties of graphene with molecular dimensions of about 10 nm or less. Graphene can also facilitate biological and medical research such as bioassay and imaging. Graphene sheets or chemically reduced graphene oxide are called GO-reduced graphene. Additionally, nanobiosensor applications and nanobiosensors have a promising role in electrochemical bases [75]. Navakul et al. proposed a new method using GO nanoparticles as nanobiosensors to rapidly detect dengue virus based on electrochemical impedance spectroscopy [76,77]. The small pieces of single-layer graphene sheets are called GQD. Additionally, GQD represents the next generation of carbon-based nanomaterials with tremendous potential in a wide range of biomedical applications [11,78,79]. GQDs are zero-dimensional materials (with the same graphene monoatomic layer) with a lateral dimension of less than 100 nm [80]. These nanomaterials allow the tracking of human cells because these nanomaterials have a robust and stable fluorescence compared to graphene and are also more hydrophobic and less toxic [78]. GQDs have a remarkable potential for delivering proteins or drug molecules to cells due to their high surface area and excellent biocompatibility [79]. In developing electrochemical biosensors, factors that make GQD useful as a surface electrode adjuster and signal booster include easy functionalization, high oxygen-rich functional groups, excellent electrochemical properties, and a large surface area relative to their volume. The electrochemical reaction of the target analyte with the modified electrode surface of the biosensor determines the electrical response; the same thing leads to the detection of analytes, which represents the purpose of these biosensors. In biomedical detection, biocompatible GQDs are used as facilitators of GQD-based nanoprobes [2,81].

The presence of several functional groups on the GQDs surface, like other graphene-based nanomaterials, allows multifaceted conjugation and conversion into ideal materials to occur for drug delivery and target-specific HIV inhibition [3,82]. Even though the study of graphene-based antivirals is still in its infancy, the QD of graphene has shown the potential to revolutionize the future of anti-HIV detection due to their tunable photoluminescence excellent colloidal stability, and acceptable biocompatibility (Figure 5) [83]. To detect HBV-DNA, Xiang et al. examined a very simple and efficient GQD-based electrochemical biosensor that acted as an electrode material. They used a safe and straightforward pyrolysis process involving citric acid to synthesize GQDs. The results showed that due to the addition of HBV-DNA, electrostatic repulsion removal caused an increase in the low values of the current peak. This increase is also due to the presence of pDNA on the surface of the modified electrode. In a focus-dependent approach where HBV-DNA-pDNA duplexes are present, this carbon-based electrochemical DNA biosensors strategy will lead to a superb linear diagnosis range between 10–500 nm and the best detection limit of 1 nm [84].

### 3.4. Carbon Nanotubes

In 1962, Clark proposed the concept of a CNT-based biosensor to describe an enzymatic electrode [85]. The biologically sensitive transducer elements are parts of a CNT-based biosensor compound. CNT acts as a biologically sensitive element with bioreceptors such as proteins, polynucleotides, and even entire biological tissues [86]. CNTs have many applications in various fields and have unique electronic, mechanical, structural, and optical properties [87]. Although spherical carbon nanomaterials have attracted attention in medicine due to their immunogenicity, biocompatibility, uniform surface chemistry, and simple geometry, their importance has decreased due to some disadvantages such as poor water solubility, lack of biodegradability, toxicity concerns, and low pK [88,89]. Additionally, since 1991, carbon nanotubes have been introduced as an excellent alternative due to their chemical stability, high heterogeneous electron transfer and long-range, high electrical conductivity, high surface area, excellent biocompatibility, and good mechanical strength in nanobiosensors [3,90,91]. Among the advantages of carbon nanotubes identified in the past decade are the fast heterogeneity electron transfer, long-range electron transfers, and high surface area, which have led to widespread use. Wiriyachaiporn et al. presented a carbon nanotagger-based lateral flow method to diagnose the influenza A virus [92]. Additionally, Li et al. used a diaminoazobenzene and multiwalled carbon nanotube-modified glassy carbon electrode to detect HBV through the electrochemical approach [93]. Since CNTs have low solubility in water and have difficulty in providing strong fluorescence in the visible region, this greatly limits their application. Therefore, CQDs as new nanomaterials based on zero-dimensional carbon, which are known for their small size and relatively strong fluorescence properties, have been considered in various applications such as biomedicine [94]. CQDs are small fluorescent carbon nanoparticles with a size that is less than 10 nm [48]. Photostability, light bleaching resistance, and nonscintillation are CQDs’ features, and QD carbon has excellent fluorescence properties compared to traditional fluorescent dyes. CQDs are safer and non-toxic at cellular and animal levels [90,91,95] and are desirable options for nanomedical applications due to the absence of visible signs of toxicity in animals, excellent water dispersion, and an average diameter of less than 10 nm [96,97]. Due to good biocompatibility, CQDs can be used as fluorescent probes for biosensing [98,99]. PL properties, small size, biocompatibility, chemically inert structure, high-temperature stability, and accessible functionalization routes are the unique features of CQDs observed in recent years [100,101]. QD carbon can be synthesized through several inexpensive and simple methods, and in vivo tracking is possible due to its excellent optical properties. Recently, suitable scaffolds that interfere with the entry of viruses into cells have been discovered for CQDs, as depicted in Figure 6 [102]. As a type of carbon nanostructure, CQDs have attractive and ideal advantages for optical sensors, including low cost-effectiveness, high water solubility, simple manufacturing methods, excellent adjustable optical properties, and chemical inertness [80,103,104]. The quantum efficiency of CQDs is low, but they have many advantages.

## 4. QD-Based Biomarkers

QD-based biomarkers are one of the potential candidates for biosensing that have attracted much attention among scientists. The detection of a specific disease or condition of interest and individual identification of a subset of the disease can be made by utilizing a QD-based diagnostic biomarker [105,106]. In addition to being a detection tool, QD-based biomarkers can be used for prognosis and predicting the outcome of the treatment of viral infections [107]. However, certain global properties such as being easy to measure, providing rapid results, non-invasive, and cost-effective are essential for all QD-based biomarkers [108]. The characteristics of QD-based biomarkers for detecting viral disease are shown in Figure 7. The benefits of QDs-based biomarkers will be more prominent when more biomarkers are analyzed [109]. Therefore, the strength and stability of probe signals will ensure efficient, rapid, and reliable identification of disease biomarkers. Additionally, in fast dynamic bio-tracking and biosensitive detection, QDs-based biomarkers can be fully integrated, with advantages such as exceptional brightness and high photostability [103,104]. QD-based biomarkers developed to identify infectious disease biomarkers are listed in Table 3. Furthermore, due to the ease of functionalization with a variety of functional groups and the high surface area to volume ratio of QD-based biomarkers, they have become a desirable option for interacting with viruses because they prevent the viruses from entering the cells [110].

## 5. Viral Infections Detected by QD-Based Biomarkers

### 5.1. Coronavirus Disease—2019

The elimination and diagnosis of SARS-CoV-2 by QDs has attracted significant interest from researchers. The initial reasons for using QDs can be attributed to their ability to be tracked under a specific wavelength of light [116]. QDs can effectively target and penetrate SARS-CoV-2 with a size between 60 and 140 nm [117]. Separation/inactivation of the S protein from SARS-CoV-2 is possible due to the positive surface charge of carbon-based QDs [118]. Furthermore, the production of reactive oxygen species within SARS-CoV-2 occurs through the interaction of the cation surface charges displayed by QDs with the negative RNA strand of the virus [119]. The antiviral effect of carbon dots (CD) against porcine reproductive and respiratory syndrome virus and pseudorabies virus was studied by Du et al. [120]. Activation of interferon-stimulated genes, especially interferon-α production, is induced by CDs that suppress viral replication. Furthermore, the integration of functional targets with QDs can effectively interact with SARS-CoV-2 entry receptors and influence genomic replication [82].

Detection of metal ions, viruses, whole cells, and hormones is performed using CNT network FET electronic biosensors. Additionally, to detect the COVID-19 virus, scientists considered CNT-FET as a reference as they took action to build a new electrochemical biosensor. In saliva samples, it is possible to utilize CNT-FET to detect SARS-CoV-2 S1 antigen digitally. Due to the development of this biosensor, SWCNTs are precipitated through non-covalent interaction using 1-pyrenbutanoic acid succinimidyl ester binder and stabilizing S1 anti-SARS-CoV-2 antibody on the SiO_2_ surface between the source-discharge channels. The scientists used a field-effect transistor of a nanotube network with a liquid gate and RNA hybridization as a signal converter and a primary signal generator, respectively. To analyze the electrical output of the CNT-FET biosensor, researchers used commercially available SARS-CoV-2 S1 antigen in developing this sensor. SARS-CoV-2 is distinct from other CoVs comprising the SARS-CoV-1 S1 or MERS-CoV S1 antigen, which can be detected accurately and sensitively using this technology. The biomarker can quickly and easily detect COVID-19 viral particles from saliva samples [121].

QDs effectively act as a biomarker and potential virus targeting agent [117]. According to the researchers, graphene-based QD biomarkers and their varieties can diagnose coronavirus; thus, they have many potential applications that can ensure the fight against COVID-19 makes significant progress (Figure 8). To target the spike protein on the coronavirus, the researchers have developed experiments combining graphene sheets that are more than 1000 times thinner than postage stamps (thickness of 0.34 nm) with antibodies. Subsequently, measurements of the atomic surface vibrations of these graphene sheets were performed when they were exposed to COVID-positive and COVID-negative samples in artificial saliva. These graphene sheets were tested in the presence of other coronaviruses such as MERS-CoV. The researchers found that the vibrations of the antibody-bound graphene sheet changed during treatment with a COVID-positive sample. However, when exposed to COVID-negative samples or other types of coronaviruses, no significant change was observed. Vibration changes were measured in less than five minutes using a Raman spectrometer. These resonant vibrations change precisely and in a quantifiable manner when a molecule such as SARS-CoV-2 interacts with graphene [122]. To identify pathogenic viruses such as SARS-CoV-2 or animal viruses that can be detected through differentiated fingerprints of their viral glycoproteins at different voltage positions, Hashemi et al. designed and reviewed a rapid electrochemical detection kit consisting of fixed screen-printing electrodes. To detect the trace of viruses in any aquatic biological environment such as blood and saliva, the researchers covered a layer of coupled graphene oxide (GO) with sensitive chemical compounds along with gold nanostars (Au NS) to activate the working electrode of the developed sensor. The results showed that this method was able to detect traces of various pathogenic viruses in about 1 min and did not require extraction and/or biomarkers to identify the target viruses. Based on the results obtained for the nanosensor, the detection limit (LOD) is 1.68 × 10^−22^ μg mL^−1,^ and the sensitivity is 0.0048 μAμg.mL^−1^. cm^−2^ [123].

### 5.2. HIV

A good alternative for ELISA kits is biomarkers, but these biomarkers have limitations such as low sensitivity. In the bioanalytical frontier, a major challenge remains the construction of multiple biosensors for the simultaneous detection of different analytes [124,125]. HIV can be detected using QD-based biomarkers by visualizing the dynamic interaction between viruses and human immune cells [126]. The presence of boronic acid on QDs as a targeting ligand can be specifically activated in the viral cycle and benefits from the better interaction of QDs with covalent, non-covalent, and electrostatic interactions, as well as their optical properties, which enable their use in HIV diagnosis [127]. Iannazzo et al. investigated the role of modifying QD-based nanomaterials in increasing their potency against HIV-1. The results also showed that modifying graphene quantum dots or QDs with non-nucleoside reverse transcriptase inhibitors (NNRTIs) could increase their activity in diagnosing HIV-1 [82]. To detect HIV dsDNA in serum, a relative fluorescence biosensor consisting of CQD and cadmium telluride quantum dots (CdTe QDs) was developed by Liang et al. CdTe QDs coated with mercapropionic acid were first coupled with mitoxantrone (MTX), a synthetic anthraquinone drug that can interfere with DNA, resulting in red fluorescence quenching at 599 nm due to electron transfer between CdTe QDs and MTX. In such a situation, only fluorescence emission at 435 nm can be detected due to the green QDs. Eventually, the CdTe QDs move further away from the MTX -ssDNA complex due to electrical repulsion. Furthermore, in the presence of HIV-dsDNA, the specific binding of MTX to dsDNA resulted in the separation between MTX and CdTe QDs Figure 9 [128].

### 5.3. HPV

A dual-color QD detection method can simultaneously detect the co-infection of HPV infection and HIV, as depicted in Figure 10 [129]. Xue et al. used the QD in situ hybridization (ISH) in studying the association of oral squamous cell carcinoma with HPV. After comparing QDISH with conventional ISH, the presence of high-risk HPV16/18 was detected. Thus, the results suggest that QD may be an effective method for detecting HPV infection and HPV-related oral squamous cell carcinoma [130]. Piao et al. also investigated different types of fluorescent nanomaterials such as metal nanoparticles, QDs, and fluorophorodized silica nanoparticles to detect human papillomavirus DNA [131]. Nejdl et al. analyzed the interactions between blue and yellow fluorescent CdS QDs to decipher the HPV-16 oncogene E6. The results showed that yellow-low fluorescent CdS-QDs showed a higher affinity for DNA (E6 HPV-16) than blue dyes [132].

### 5.4. Hepatitis

Recently, QD-based biomarkers have been introduced, where the unique features of QDs allow them to be used to detect hepatitis viruses. They do not require sample preparation compared to traditional strategies [133]. The detection of specific targets (such as hepatitis) by using QD-based biomarkers is done through the interaction of biological molecules such as the substrate-enzyme reaction, antigen-antibody, or ligand-receptor complexes [134].

The electro-quantitative luminescence method is also used to detect hepatitis C and hepatitis B virus using CdTe multicolor QDs (Figure 11). Ideally, the detection limit for hepatitis B virus is 0.082 picomoles per liter and for hepatitis C virus is 1.341 picomoles per liter. The DNA sensor has the appropriate specificity, sensitivity, stability, and reproducibility. This sensor is used to detect hepatitis C virus target RNA and HBV target DNA in humans, and the results have been evaluated to be satisfactory [135]. In clinical diagnosis, sensitive and convenient detection of HBV gene mutations is essential. According to Zhang et al., to detect mutations in the HBV gene in real serum samples of patients with chronic hepatitis B (CHB) who had received antiviral therapy with lamivudine or telbivudine, a sensitive, low-cost, and convenient QDs-mediated fluorescent method was developed [136]. Chowdhury et al. investigated an electrochemical sensor based on enhanced QDs to detect the hepatitis E virus (HEV). The results showed that the proposed sensor based on QDs could pave the way for developing robust and powerful assay methods for detecting HEV [137]. Liu et al. investigated a protein array based on QDs encoded microbes for the diagnosis of hepatitis C virus. To evaluate the protein array based on QDs encoded microbes, 120 HCV-positive and 50 HCV-negative samples were tested and compared with the results of the recombinant immunoblot assay (RIBA) as the gold standard [138]. Wang et al. investigated the highly efficient production of QD barcodes at multiple scales to diagnose multiplex hepatitis B. Five HBV indicators were selected as diagnostic targets to examine the feasibility of QD barcodes in high-performance detection. The results showed that these QD barcodes are very useful for the simultaneous and selective detection of many different biomolecular targets [139].

### 5.5. Dengue Virus, Influenza Virus, Zika Virus, and Norovirus

QDs are used for dengue virus, influenza virus, Zika virus, and norovirus. Current methods of diagnosing viruses such as dengue, influenza, Zika, and norovirus can take two to six days, delaying effective treatment. This is one of the problems with current diagnostic technologies, which take too long. Recent diagnostic tests take up to five days to determine if a virus is present and another day or more to determine which virus it is. Therefore, GQDs and CQDs can be used with advanced technology made of multi-colored and microscopic fluorescent beads that attach to molecular structures unique to the viral envelope and the cells they infect. Thanks to their high sensitivity, quantum dot systems also enable the detection of viral particles during infection within a few hours, instead of the two to five days required by current tests. Quantum dots are not only cost-effective but can also detect at least four major respiratory viruses simultaneously, including dengue, influenza, Zika, and norovirus. Colored quantum dots are attached to different “linker” molecules that bind to the surface structures of the different viruses. Quantum dots are available in dozens of colors, while antibodies specific to the four respiratory viruses have been identified and can be used as linker molecules [140,141].

## 6. Multiplex Detection of Viral Infection

However, detecting a single biomarker is insufficient to diagnose a disease, as a single biomarker may indicate more than one disease [142]. For example, using one biomarker to detect viral particles in infected people can lead to false-positive and negative results [143]. Multiple detections of biomarkers in a single assay have also been used to increase detection accuracy by providing more accurate scientific information and improve detection performance by delivering faster analysis to prevent misdiagnosis [144]. Therefore, in clinical detection, it is crucial to develop techniques for multiple detections of disease biomarkers [145]. Since multiple biomarkers play a role in disease development and progression, practical and sensitive detection methods capable of simultaneously detecting multiple biomarkers associated with a given disease (i.e., multiple detections) are urgently needed. The advantages of biomarker multiplex detection include lowering the cost of detecting a specific infection and reducing patient pain due to low sample consumption [146]. In recent years, biomarker multiplex detection has attracted much attention for the reasons mentioned above.

Nanoflares [147], SiO_2_-encoded porous photonic crystal disks [148], chemiluminescence [149], silicon nanowires [150], QDs [151], and fluorescent nanorods [152] are among the various multiple biomarker detection techniques. Additionally, QD-based biomarkers are used for multiple and selective detections of viruses, although optimization of many new methods in assay conditions is necessary [151]. Thus, by adjusting the chemical composition and size of QDs, QD-based biomarkers with different emission wavelengths can be obtained. They are easily distinguished, identified, and excited by a single light source, among the many advantages of QD-based biomarkers in multiple detections [153]. Table 4 shows the QD-based biomarkers for multiple diagnoses of viral infections.

## 7. Conclusions and Perspective

Early detection of viral infections and their effective treatment is key to reducing them and improving treatment success rates and best chances. Molecular-based tests are not very sensitive, require a long time to perform, and do not allow on-site detection of viral infections in biological media. Currently, the majority of routine viral diagnoses are based on molecular tests. Quantum dot biomarkers, which enable miniaturized biosensors for effective early diagnosis of viral infections in the field, have shown excellent potential for revolutionizing viral infection. Strong covalent binding between these molecules has led to advanced sensor substrates that can be used for precise detection of viral biomarkers and monitoring treatment progress at the point of care. This review describes different approaches for synthesizing and functionalizing QDs by incorporating various biomolecules that can recognize and convert them into signals, such as antigens, enzymes, hormones, proteins, and virus-associated by-product biomolecules on the surface of viruses. Multiplex biosensors have been developed to study the effectiveness of virus detection systems. The results of these tests have shown that these tools help diagnose infections and evaluate the efficacy of viral surveillance, indicating their potential use in the clinical setting to detect and treat viral infections.

Although the multiplex biosensors developed have excellent performance, future research in this area must take into account that integrated biosensors can be used as part of point-of-care systems to detect multiple biomarkers simultaneously. In addition, laboratory instruments should provide accuracy and reliability while being sensitive, fast, and convenient to use. Developing QD biomarkers or biosensors is a significant challenge due to the difficulty of synthesizing high-quality and stable QDs with defined size, shape, charge, and degree of agglomeration. These properties significantly impact the physicochemical properties and performance of the QD biomarkers. Further studies are needed to explore the outstanding results obtained with these nanomaterials for biosensing and compare the results obtained with those of other existing techniques. A new protocol for detecting viral biomarkers with these highly sensitive biosensors needs to be developed. With the proper knowledge from biology, physics, chemistry, and engineering, advanced QD-based biomarkers will enable simple, rapid, and accurate early and multiplex diagnosis of various viral infections in the clinic in the near future.

## Abbreviations

QDsQuantum DotsNWsNanowiresGOGraphene OxideGQDGraphene Quantum DotsCNTCarbon nanotubeCQDsCarbon Quantum DotSWCNTsSingle-Walled Carbon NanotubesCoVCoronavirusFETField-Effect TransistorSARS-CoVSevere Acute Respiratory Syndrome CoronavirusMERSMiddle East Respiratory SyndromeHIVHuman Immunodeficiency VirusAIDSAcquired Immune Deficiency SyndromeHPVHuman papillomavirusPCRPolymerase chain reaction

## Figures and Tables

**Figure 1 pharmaceuticals-15-00880-f001:**
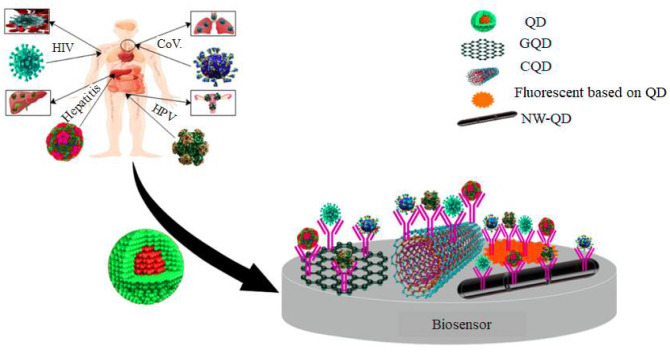
QD-based nanosensors to detect viruses.

**Figure 2 pharmaceuticals-15-00880-f002:**
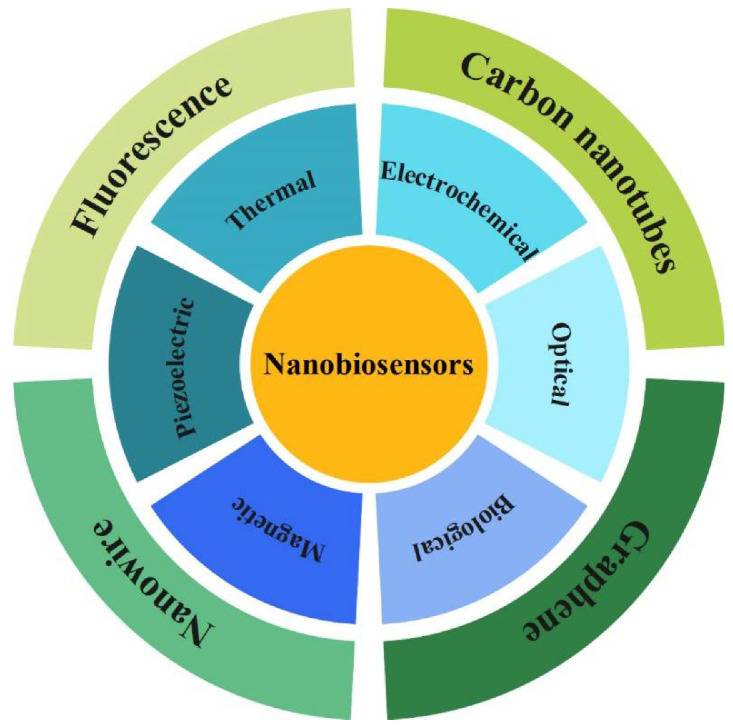
Classification of nanobiosensors for viral detection.

**Figure 3 pharmaceuticals-15-00880-f003:**
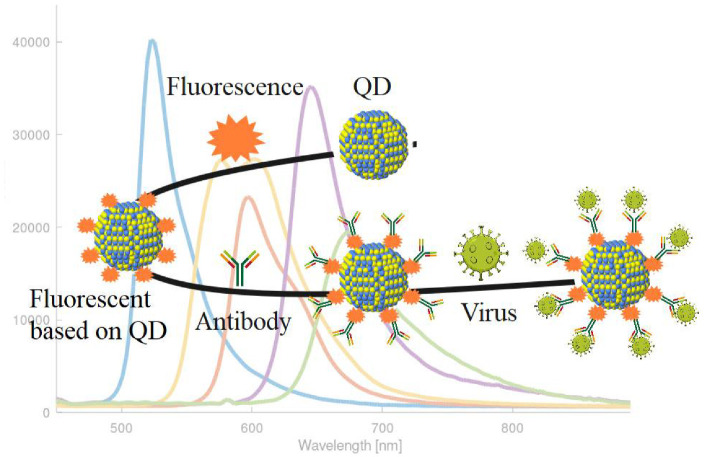
Fluorescence-based on QDs for detecting pandemic viruses.

**Figure 4 pharmaceuticals-15-00880-f004:**
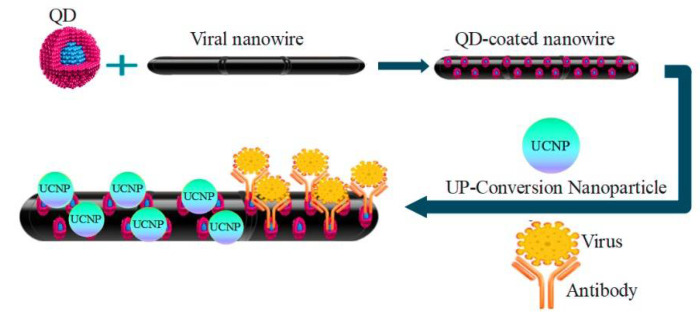
QD-coated viral nanowire for detecting pandemic viruses.

**Figure 5 pharmaceuticals-15-00880-f005:**
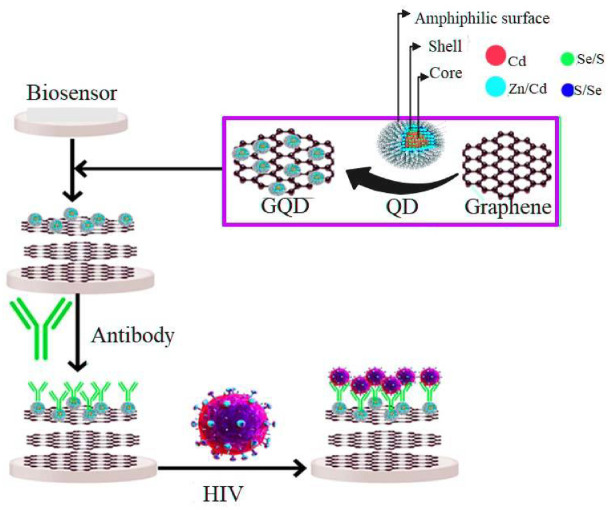
The GQD for detecting HIV.

**Figure 6 pharmaceuticals-15-00880-f006:**
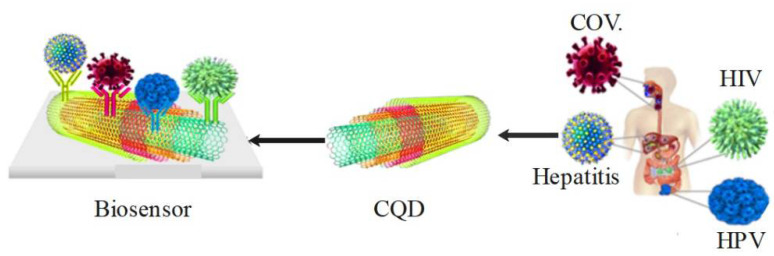
QD carbon nanotubes detect pandemic viruses.

**Figure 7 pharmaceuticals-15-00880-f007:**
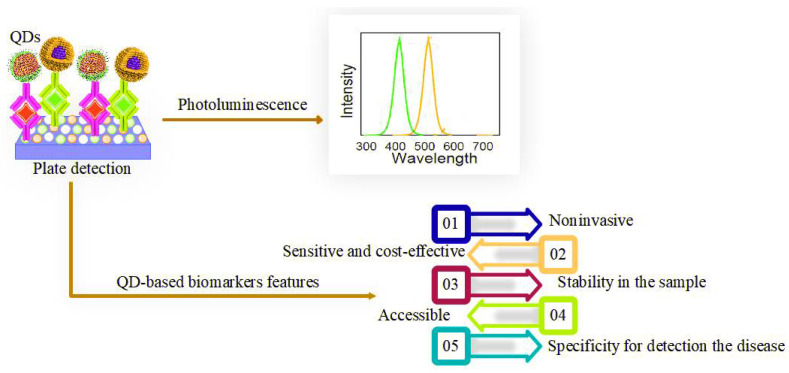
QD-based biomarker features for the detection of viral disease.

**Figure 8 pharmaceuticals-15-00880-f008:**
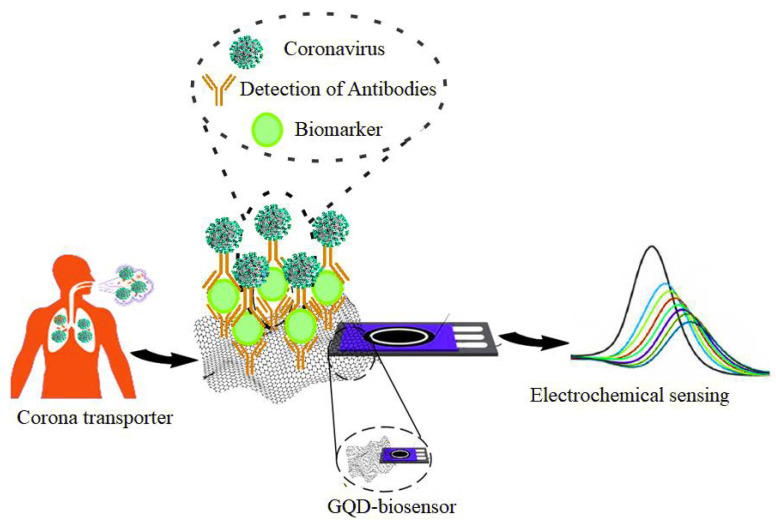
Detection of coronaviruses by using QD-based graphene biomarker.

**Figure 9 pharmaceuticals-15-00880-f009:**
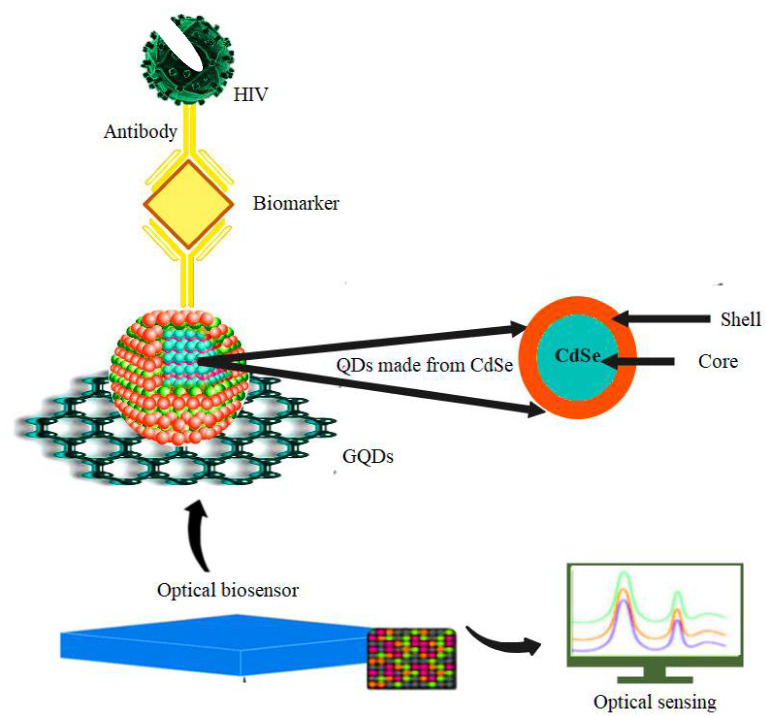
Detection of HIV by using QD-based biomarkers made from CdSe with optical sensors.

**Figure 10 pharmaceuticals-15-00880-f010:**
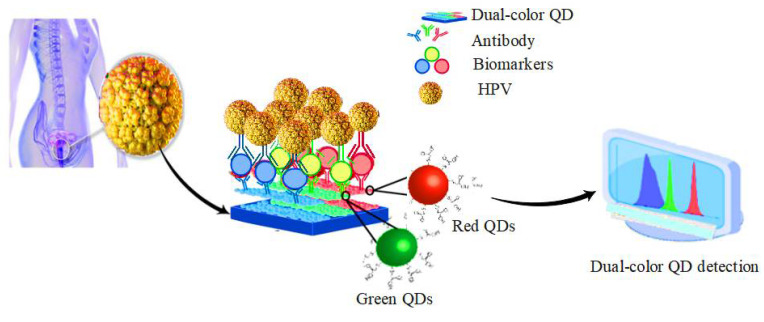
Detection of HPV infection by using dual-color QD detection method.

**Figure 11 pharmaceuticals-15-00880-f011:**
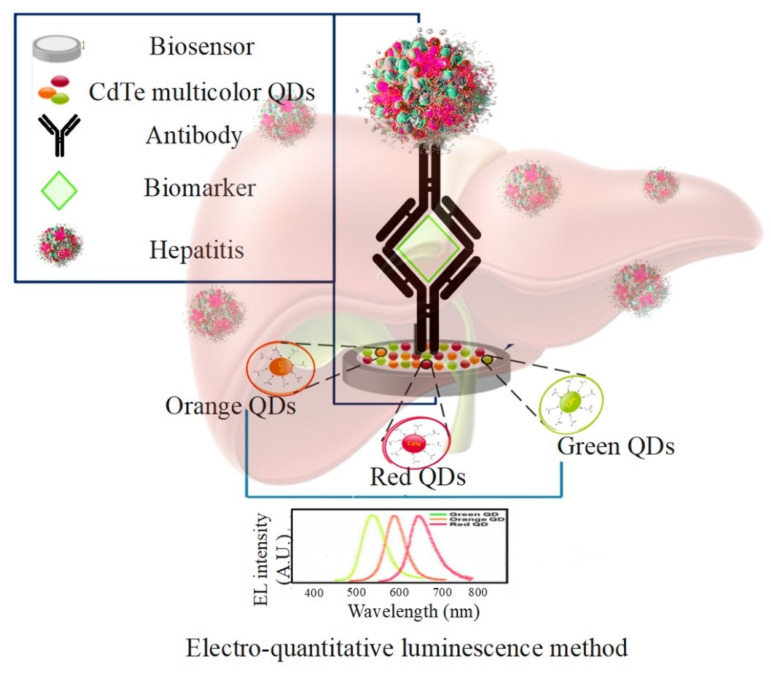
Detection of hepatitis C and hepatitis B virus using CdTe multicolor QDs and gold nanoparticles in the electro-quantitative luminescence method.

**Table 2 pharmaceuticals-15-00880-t002:** Advantages and disadvantages of semiconductor, carbon, and graphene quantum dots.

	Advantages	Disadvantages	Ref.
Semiconductor quantum dots	High photostability Resistance to photobleaching The narrow and symmetric peak of emission Stokes shift is more than 200 nm (ease to detection) High quantum yield Long lifetime Resistance to chemical and biological degradation	The multiexponential decline of fluorescence and blinking of separate QDs The high background level of deduction and accumulation of QDs in the reticuloendothelial system Incomplete elimination of QDs after injection into an organism The high toxicity of QDs when used in in vivo systems	[38,39]
Carbon quantum dots	Non-toxic Cost-effective Eco-friendly Simple Cheap Facile Rapid Scalable Size and nanostructure are controllable	Toxic acid/base reagents Broad size distribution Expensive oxidants Long synthesis duration	[40,41]
Graphene quantum dots	Simple and effective Environmentally friendly Shorten the reaction time Improve the production yield High levels of stability Uniform size distribution	Some strong oxidizers cause burning or explosion The carbon materials need to be treated through strong oxidation before the reactions happen Difficult to realize mass production because of the low product yield	[42,43,44,45]

**Table 3 pharmaceuticals-15-00880-t003:** QD-based biomarkers that are made to detect biomarkers of infectious diseases.

Infectious Disease	Infectious Biomarker	Detection Techniques	Ref.
Hepatitis B	HBV virus	Microfluidic device with microbead array and QD	[111]
HIV	Anti-HIV antibody	Biosensors	[112]
Hepatitis C	Anti-HCV antibodies	Optical immunosensors	[113]
Autoimmune hepatitis	Serum levels: IL-6, IL-8, IL-17, IL-21, tumor necrosis factor (TNF)-α	enzyme-linked immunosorbent assay	[114]
Hepatitis B	Hepatitis B surface antibodies	Surface acoustic wave immunosensor	[115]

**Table 4 pharmaceuticals-15-00880-t004:** QD-based biomarkers in multiplex detection of viral infections.

Biomarkers	Detection Time	Analysis Mode	Detection Method	Ref.
CEA and NSE	<15 min	Quantitative	Fluorescent detection	[154]
Myo, cTnI, and CKMB	17 min	Quantitative	SERS detection	[155]
PSA and EphA2	Not mentioned	Quantitative	Fluorescent detection	[156]
BoNT-A, BoNT-B, and BoNT-E	25 min	Quantitative	Magnetic detection	[157]
SD, TC, and CT	10 min	Semiquantitative	Colorimetric detection	[158]
DENV NS1 and ZIKV NS1	Not mentioned	Quantitative	SERS detection	[159]
AFP and CEA	30 min	Semiquantitative	Colorimetric detection	[160]
Myo, cTnI, and CKMB	45 min	Quantitative	SERS detection	[161]
MOP, fentanyl, and MET	<20 min	Quantitative	Magnetic detection	[162]

## Data Availability

All data generated or analyzed during this study are included in this published article.

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
