# Peer review of "The Pivotal Role of Quantum Dots-Based Biomarkers Integrated with Ultra-Sensitive Probes for Multiplex Detection of Human Viral Infections"

_pharmaceuticals, 2022, doi:10.3390/ph15070880_

Round 1
Reviewer 1 Report
1. Sections on the chemistry of semiconductor QDs, carbon QDs, and graphene QDs and their functionalization strategies must be provided.
2. A section describing the comparative merits and demerits of semiconductor QDs, carbon QDs, and graphene QDs must be provided.
3. Sections on the use of QDs for Dengue virus, influenza virus, Zika virus and norovirus must be provided
4. The following relevant paper must be cited:
QDs chemistry: Materials Science in Semiconductor Processing 90 (2019) 162–170
Author Response
Dear reviewer,
Pharmaceuticals journal
Thank you for considering our manuscript. We would like to appreciate for appending the reviewers' comments. We have carefully reviewed the comments and have revised the manuscript, accordingly. Our responses are given in a point-by-point manner below. Changes in the manuscript has been highlighted in yellow font color.
We hope the revised version is now suitable for publication and look forward to hearing from you in due course.
Sincerely,
Dr. Ahmad Gholami
on behalf of all authors
Reviewer 1.
- Sections on the chemistry of semiconductor QDs, carbon QDs, and graphene QDs and their functionalization strategies must be provided.
Response: Thank you very much for your valuable suggestion. This is really a good comment at a high level. We have included this section in the article.
- the chemistry of semiconductor QDs, carbon QDs and graphene QDs and their functionalization strategies.
A quantum dot is a region in a semiconductor crystal that includes electrons, holes, or both in three dimensions. All three dimensions of matter are on the nanometer scale. The main feature of these dots is light scattering. Their dimensions are so small that the properties of matter cannot be explained by the laws of classical physics; only quantum physics can explain the behavior of matter. Quantum dots are a unique class of semiconductors because of their small size. The importance of quantum dots as semiconductor gates is that the electrical conductivity of these materials can be changed by external stimuli, such as electric fields or light radiation, to the extent that they switch from being nonconducting to conducting and act as switches. Compared to real atoms, semiconductor quantum dots have the special feature that the number of free electrons can be changed using external instruments [46-49]. This is a simple way to create artificial atoms with 3, 2, 1 or more electrons. Therefore, adding or subtracting electrons to quantum particles will lead to a wide range of synthetic materials. However, it must be remembered that the synthetic materials produced by this method will not have all the properties of the original materials. It is likely that the materials produced will ultimately consist of nanorobots that will be under human command in appearance and function. Therefore, the two properties of quantum confinement effects and edge effects are the fundamental properties of graphene quantum dots and carbon quantum dots, which have unique chemical and physical properties such as non-toxicity, biocompatibility, stable optical properties, and so on. One of the results of these fundamental properties is that they can be used in many applications in various fields such as detectors, medicine, optoelectric devices, batteries, supercapacitors, catalysts, and sensors [50-53]. Research on graphene quantum dots and carbon quantum dots is still in its infancy, and many challenges remain to be solved. Graphene quantum dots were first fabricated by Geim and Ponomarenko in 2008. Graphene quantum dots are two-dimensional structures with a graphene lattice that are less than 10 layers thick and less than a few tens of nanometers in size. These structures have many novel properties, including a unique luminosity due to the effects of quantum confinement [54-56]. Carbon quantum dots were first produced in 2004 in an attempt to purify single-walled carbon nanotubes. Carbon quantum dots have a spherical morphology of less than 20 nm. Later, in 2006, the above structure was also successfully fabricated by laser ablation of graphite powder, thus taking a new step towards the identification and use of more of these materials [57, 58]. Graphene quantum dots and pure carbon quantum dots have many limitations that limit their applications. To extend their applications to different fields, these structures can be functionalized by various methods, such as doping with different atoms, forming composites with minerals or polymers, controlling the size and changing the morphology, and optimizing them for specific applications by changing their chemical, optical, and electronic properties. Doping of semiconductor materials is a fundamental process in the semiconductor industry because it can change the key physical, chemical, and electronic properties of the materials and improve their performance in various applications. Graphene quantum dots and carbon quantum dots are no exception to this rule, as their various properties can be improved in this way. So far, most phosphorus, nitrogen, sulfur, selenium, chlorine, fluorine, and boron atoms have been used to dope the aforementioned quantum dots, and the results have been studied using various approaches, such as increasing the band gap and shifting the optical absorption peak, as well as changing the intensity of the photoluminescence emission. Another strategy is to change the optical, physical, and chemical properties of graphene quantum dots and carbon quantum dots by controlling the size and shape of these particles. In addition, there are various methods in which the fabrication conditions, pH factor, etc. are changed to alter the size and morphology of graphene quantum dots. These methods have resulted in shifting the band gap from the blue region of the spectrum to the red region [59-62]. - Rogach, A.L., Semiconductor nanocrystal quantum dots. Verlag: Wien, 2008.
- Zhou, W. and J.J. Coleman, Semiconductor quantum dots. Current Opinion in Solid State and Materials Science, 2016. 20(6): p. 352-360.
- Adegoke, O., et al., Passivating effect of ternary alloyed AgZnSe shell layer on the structural and luminescent properties of CdS quantum dots. Materials Science in Semiconductor Processing, 2019. 90: p. 162-170.
- Hashemi, S.A. and S.M. Mousavi, Effect of bubble based degradation on the physical properties of Single Wall Carbon Nanotube/Epoxy Resin composite and new approach in bubbles reduction. Composites Part A: Applied Science and Manufacturing, 2016. 90: p. 457-469.
- Harrison, P. and A. Valavanis, Quantum wells, wires and dots: theoretical and computational physics of semiconductor nanostructures. 2016: John Wiley & Sons.
- Gerion, D., et al., Synthesis and properties of biocompatible water-soluble silica-coated CdSe/ZnS semiconductor quantum dots. The Journal of Physical Chemistry B, 2001. 105(37): p. 8861-8871.
- Vasilevskii, M., et al., Effect of size dispersion on the optical absorption of an ensemble of semiconductor quantum dots. Semiconductors, 1998. 32(11): p. 1229-1233.
- Mousavi, S.M., et al., Modification of phenol novolac epoxy resin and unsaturated polyester using sasobit and silica nanoparticles. Polymers from Renewable Resources, 2017. 8(3): p. 117-132.
- Ponomarenko, L.A., et al., Chaotic Dirac billiard in graphene quantum dots. Science, 2008. 320(5874): p. 356-358.
- Mousavi, S., M. Zarei, and S. Hashemi, Polydopamine for biomedical application and drug delivery system. Med Chem (Los Angeles), 2018. 8: p. 218-29.
- Amani, A.M., et al., Electric field induced alignment of carbon nanotubes: methodology and outcomes, in Carbon nanotubes-recent progress. 2017, IntechOpen.
- Wang, Y. and A. Hu, Carbon quantum dots: synthesis, properties and applications. Journal of Materials Chemistry C, 2014. 2(34): p. 6921-6939.
- Mousavi, S.M., et al., Synthesis of Fe3O4 nanoparticles modified by oak shell for treatment of wastewater containing Ni (II). Acta Chimica Slovenica, 2018. 65(3): p. 750-756.
- Bezzon, V.D., et al., Carbon nanostructure-based sensors: a brief review on recent advances. Advances in Materials Science and Engineering, 2019. 2019.
- Mansuriya, B.D. and Z. Altintas, Carbon Dots: Classification, Properties, Synthesis, Characterization, and Applications in Health Care—An Updated Review (2018–2021). Nanomaterials, 2021. 11(10): p. 2525.
- Xia, C., et al., Evolution and synthesis of carbon dots: from carbon dots to carbonized polymer dots. Advanced Science, 2019. 6(23): p. 1901316.
- Mousavi, S., et al., Improved morphology and properties of nanocomposites, linear low density polyethylene, ethylene-co-vinyl acetate and nano clay particles by electron beam. Polymers from Renewable Resources, 2016. 7(4): p. 135-153.
- A section describing the comparative merits and demerits of semiconductor QDs, carbon QDs, and graphene QDs must be provided.
Response: Thank you very much for your valuable suggestion. This comment is outstanding at a high level. We have included this section in the article as follows:
2.1 Comparing the advantages and disadvantages of semiconductor QDs, carbon QDs, and graphene QDs.
For quantum dots made of semiconductor materials, the concepts related to capacitance, conductivity, and the forbidden band for semiconductor materials also apply to quantum dots. Nevertheless, a critical difference between them is that electron transfer occurs easily in bulk materials because there is enough space for the electrons to move. In quantum dots, however, this electron transfer occurs at a minimal radial distance called the "Bohr radius". When the dimensions of the quantum dots or quantum crystals are as small as the Bohr radius, the electron can no longer move as easily in matter, and the laws of electron movement and transfer change dramatically. This leads to unique optical properties, including the effect of light absorption and reflection in semiconductor crystals with dimensions that are in the range of the Bohr radius. Another feature of semiconductor quantum dots is that a change in the number of atoms causes a change in the forbidden band; in other words, it changes the energy difference between the conducting layer and the capacitance. In addition to the number of atoms, the way they are arranged at the quantum dot level also affects the magnitude of the energy difference [63-65]. Carbon quantum dots (CQDs) and graphene quantum dots are a new class of carbon nanomaterials that have recently emerged and are considered as potential competitors of semiconductor quantum dots due to their low toxicity, environmental friendliness and low fabrication cost. Although many important and practical advantages have been identified for the above structures, further research is still being conducted to improve the recognition of these advantages and their use in various applications. To meet the needs of industry, mass production of graphene and carbon quantum dots at relatively low cost is required. Of course, there are challenges to be overcome in their industrialisation and mass production. The reported quantum efficiency of quantum dots is lower than that of conventional semiconductor quantum dots. Therefore, the low quantum efficiency is another challenge that requires further research. Another problem is the accuracy of the results of optical studies, which have always been controversial, and only changing the fabrication method drastically changes the comparison of the results. This has led to a lack of understanding of the mechanism of photoluminescence and the associated analysis of graphene and carbon quantum dots [66-69]. The following table shows the advantages and disadvantages of semiconductor, carbon and graphene quantum dots.Table 2. Advantages and disadvantages of semiconductor, carbon and graphene quantum dots
|
|
Advantages |
Disadvantages |
Ref. |
|
|
semiconductor quantum dots |
|
|
[70-73] |
|
|
Carbon quantum dots |
|
|
[74-79] |
|
|
Graphene quantum dots |
|
|
|
[80-83]
|
- Vanmaekelbergh, D. and P. Liljeroth, Electron-conducting quantum dot solids: novel materials based on colloidal semiconductor nanocrystals. Chemical Society Reviews, 2005. 34(4): p. 299-312.
- Yoffe, A.D., Semiconductor quantum dots and related systems: electronic, optical, luminescence and related properties of low dimensional systems. Advances in physics, 2001. 50(1): p. 1-208.
- Reshma, V. and P. Mohanan, Quantum dots: Applications and safety consequences. Journal of Luminescence, 2019. 205: p. 287-298.
- Singh, I., et al., Carbon quantum dots: Synthesis, characterization and biomedical applications. Turkish Journal of Pharmaceutical Sciences, 2018. 15(2): p. 219.
- Hu, Y., et al., Waste frying oil as a precursor for one-step synthesis of sulfur-doped carbon dots with pH-sensitive photoluminescence. Carbon, 2014. 77: p. 775-782.
- Dager, A., et al., Synthesis and characterization of mono-disperse carbon quantum dots from fennel seeds: photoluminescence analysis using machine learning. Scientific reports, 2019. 9(1): p. 1-12.
- Tian, P., et al., Graphene quantum dots from chemistry to applications. Materials today chemistry, 2018. 10: p. 221-258.
- Barroso, M.M., Quantum dots in cell biology. Journal of Histochemistry & Cytochemistry, 2011. 59(3): p. 237-251.
- Wang, L., et al., Semiconducting quantum dots: Modification and applications in biomedical science. Science China Materials, 2020. 63(9): p. 1631-1650.
- Jin, Z. and N. Hildebrandt, Semiconductor quantum dots for in vitro diagnostics and cellular imaging. Trends in biotechnology, 2012. 30(7): p. 394-403.
- Pleskova, S., E. Mikheeva, and E. Gornostaeva, Using of quantum dots in biology and medicine. Cellular and molecular toxicology of nanoparticles, 2018: p. 323-334.
- Shen, L., et al., The production of pH-sensitive photoluminescent carbon nanoparticles by the carbonization of polyethylenimine and their use for bioimaging. Carbon, 2013. 55: p. 343-349.
- Deng, J., et al., Electrochemical synthesis of carbon nanodots directly from alcohols. Chemistry–A European Journal, 2014. 20(17): p. 4993-4999.
- Li, X., et al., Preparation of carbon quantum dots with tunable photoluminescence by rapid laser passivation in ordinary organic solvents. Chemical Communications, 2010. 47(3): p. 932-934.
- Liu, Y., et al., One-step microwave-assisted polyol synthesis of green luminescent carbon dots as optical nanoprobes. Carbon, 2014. 68: p. 258-264.
- Farshbaf, M., et al., Carbon quantum dots: recent progresses on synthesis, surface modification and applications. Artificial cells, nanomedicine, and biotechnology, 2018. 46(7): p. 1331-1348.
- Desmond, L.J., A.N. Phan, and P. Gentile, Critical overview on the green synthesis of carbon quantum dots and their application for cancer therapy. Environmental Science: Nano, 2021. 8(4): p. 848-862.
- Chen, W., et al., Synthesis and applications of graphene quantum dots: a review. Nanotechnology Reviews, 2018. 7(2): p. 157-185.
- Campuzano, S., P. Yáñez-Sedeño, and J.M. Pingarrón, Carbon dots and graphene quantum dots in electrochemical biosensing. Nanomaterials, 2019. 9(4): p. 634.
- Kundu, S. and V.K. Pillai, Synthesis and characterization of graphene quantum dots. Physical Sciences Reviews, 2020. 5(4).
- Shen, J., et al., A critical review of graphene quantum dots: synthesis and application in biosensors. Nano, 2021. 16(01): p. 2130001.
- Sections on the use of QDs for Dengue virus, influenza virus, Zika virus and norovirus must be provided
Response: Thank you for your great comment. We have added this section to the article as follows:
5.5 The use of QDs for dengue virus, influenza virus, Zika virus and norovirus
Current methods of diagnosing viruses such as dengue virus, influenza virus, Zika virus and norovirus can take two to six days, delaying effective treatment. This is one of the problems with current diagnostic technologies, which take too long. Current diagnostic tests take up to five days to determine if a virus is present and another day or more to determine which virus it is. Therefore, a new method called Quan-tum Dots can be used with advanced technology made of multi-coloured and microscopic fluorescent beads that attach to molecular structures unique to the viral envelope and the cells they infect. Thanks to their high sensitivity, quantum dot systems also enable the detection of viral particles during infection within a few hours, instead of the two to five days required by current tests. Quantum dots are not only cost-effective, but can also detect at least four major respiratory viruses simultaneously, including dengue virus, influenza virus, Zika virus and norovirus. Coloured quantum dots are attached to different "linker" molecules that bind to the surface structures of the different viruses. Quantum dots are available in dozens of different colours, while antibodies specific to the four respiratory viruses have been identified and can be used as linker molecules [84-86].
- Modani, S., M. Kharwade, and M. Nijhawan, Quantum dots: a novelty of medical field with multiple applications. Int J Curr Pharm Res, 2013. 5(4): p. 55-59.
- Tope, S., et al., Therapeutic application of quantum dots (QD). The Pharma Innovation, 2014. 2(12, Part A): p. 86.
- Salisbury, D.F., Quantum dots detect viral infections. PHYSorg. com, 2005. 10.
- The following relevant paper must be cited:
QDs chemistry: Materials Science in Semiconductor Processing 90 (2019) 162–170
Response: This article was carefully studied and enriched the text as reference no. [48].
- Adegoke, O., et al., Passivating effect of ternary alloyed AgZnSe shell layer on the structural and luminescent properties of CdS quantum dots. Materials Science in Semiconductor Processing, 2019. 90: p. 162-170.
Reviewer 2 Report
This review deals with the use of quantum dots for the detection of pathogens, with special emphasis on viruses such as HIV, Hepatitis for SAR-CoV-2. Although the title would point out that the focus of the article is on multiplexing, mostly a not sufficiently specific part at the end of the article deals with it. The article should be fully re-written to improve English writing style, language and scientific clarity before a potential re-submission. More on that and other points of attention in the following specific comments. Graphical material is very approximatively put together, including stretched out of proportion, pixelated, badly rotated copy-pasted components. All this material should be carefully revised. Free tools are available to produce professional-quality graphics for publication in a few hours. Please make use of these tools and cite relevant sources when copying graphical content from the web or articles in the figure captions.
Many sentences are very unclear, or scientific terms are replaced by different, inadequate words, and should be re-written with the help of native or fluent English-speaking editor, for example:
“Since fluorescence-based techniques are susceptible, it can be said that the use of fluorescent nanoparticles as a probe for bioanalytical applications is an up-and-coming technique”
“By transforming the module of sensing materials-based analytical instruments at the nanoscale level”
“Biomarkers are the purposes that these devices used to diagnose and describe disease status in real time”
“These benefits can greatly improve the sensitivity of biological detection and imaging by at least one to two times” you mean by one to two orders of magnitude? One to two times would not be a great selling point.
“Fluorescent biosensors can be used to detect viruses due to various parameters such as intensity, energy
transfer, longevity, and quantum efficiency” by longevity do you mean lifetime?
A highly ultrasensitive fluorescence sensor should be developed for detecting the appropriate fluorescence molecules, laser power regulation, and virus detection” I don’t see what you mean with laser power regulation.
“The optical properties of graphene and GO are of interest and somewhat unknown.” Please be more specific.
“their importance has reduced due to some disadvantages” please be specific.
Enumeration starting line 269 should be reworked, or turned into a clear sentence. A few words should be sufficient to say the QDs should be affordable, enable highly sensitive assays and not display toxicity.
“QD-based biomarkers can be categorized into four groups on the basis of their contribution to business, regulatory, and clinical decision-making” I really don’t see where the authors want to go with that opening sentence. At the end of the paragraph, four categories are mentioned again: “Figure 8 shows the classification of QD-based biomarkers into four categories: target, mechanism, pathophysiological, and detection. Could these categories be clarified or corrected if needed? The flow chart and explanation is likely to match general/textbook medical/pharmaceutical knowledge of the positioning of drugs and/or diagnostics in patient care, with not much specificity related to QDs, so this part and figure could be removed.
“To target the infamous spike protein on the coronavirus,” infamous could be removed remain objective.
“researchers designed experiments in which graphene sheets, which are more than 1,000 times thinner than postage stamps,” The authors should provide precise measurements and units.
“QDs are paired with many biological molecules such as proteins, drugs, DNA probes, and oligonucleotide probes due to the modifiable nature of semiconductor nanocrystals with regard to their surface.”, “The ability of QDs to quickly change their level as well as their faster and cheaper detection mode make them a successful tool for any diagnosis” similar statements are repeated many times in the article. Once or twice in the text is sufficient, then focus should be made on specific information. The authors should try to avoid too many repetitions of basic statements, in particular when describing analytical approaches to detect different pathogens.
All the pathogens cited are well known and extensively reviewed, and the focus of this review is on QDs, so the pathogens/medical conditions should be more briefly described, and the readers referred to specialised reviews.
The conclusion and perspective gives mostly very generic statements with low added value. This should be reworked to extract the most interesting, specific information and take home messages from the assembled literature, to give the readers the feeling to have learned something they do not already know, or a perspective they did not yet experience.
Author Response
Dear reviewer,
Pharmaceuticals journal
Thank you for considering our manuscript. We would like to appreciate for appending the reviewers' comments. We have carefully reviewed the comments and have revised the manuscript, accordingly. Our responses are given in a point-by-point manner below. Changes in the manuscript has been highlighted in yellow font color.
We hope the revised version is now suitable for publication and look forward to hearing from you in due course.
Sincerely,
Dr. Ahmad Gholami
on behalf of all authors
Reviewer 2
This review deals with the use of quantum dots for the detection of pathogens, with special emphasis on viruses such as HIV, Hepatitis for SAR-CoV-2. Although the title would point out that the focus of the article is on multiplexing, mostly a not sufficiently specific part at the end of the article deals with it.
- The article should be fully re-written to improve English writing style, language and scientific clarity before a potential re-submission.
Response: Thank you for your consideration and your great comments. The full text of the manuscript has been revised by a professional scholar. The overall writing style, language and scholarship were carefully reviewed again for further editing.
- More on that and other points of attention in the following specific comments. Graphical material is very approximatively put together, including stretched out of proportion, pixelated, badly rotated copy-pasted components. All this material should be carefully revised. Free tools are available to produce professional-quality graphics for publication in a few hours. Please make use of these tools and cite relevant sources when copying graphical content from the web or articles in the figure captions.
Response: Thank you for your valuable suggestion. We have tried to revise all the figures as far as possible. As our team includes a professional scientific graphic designer, we have not used free graphic tools.
- Many sentences are very unclear, or scientific terms are replaced by different, inadequate words, and should be re-written with the help of native or fluent English-speaking editor.
Response: Thank you very much in advance for your valuable review. The overall writing style, language and sentences was rechecked carefully and unclear sentences, inadequate scientific terms and words was re-write point by point. For example, the mentioned sentence was revised as follow:
For example:
“Since fluorescence-based techniques are susceptible, it can be said that the use of fluorescent nanoparticles as a probe for bioanalytical applications is an up-and-coming technique”
“The use of fluorescent nanomaterials as probes for bioanalytical applications is a new and emerging technique due to their high stability, sensitivity, large band gap, superior water solubility, and excellent fluorescence properties.”
“By transforming the module of sensing materials-based analytical instruments at the nanoscale level”
Nanoscale analytical tools are having a remarkable impact on transforming current analytical methods into diagnostic approaches by transforming their sensing module for the detection of biological molecules such as viruses (cold, flu, chickenpox, HIV, SARS-CoV-2).
“Biomarkers are the purposes that these devices used to diagnose and describe disease status in real time”
Biomarkers are essential for diagnosing, predicting, and monitoring disease progression and drug development in nanobiosensor devices.
“These benefits can greatly improve the sensitivity of biological detection and imaging by at least one to two times” you mean by one to two orders of magnitude? One to two times would not be a great selling point.
These advantages could improve biological detection and imaging sensitivity by at least one to two orders of magnitude.
“Fluorescent biosensors can be used to detect viruses due to various parameters such as intensity, energy transfer, longevity, and quantum efficiency” by longevity do you mean lifetime?
Fluorescent biosensors can detect viruses due to various parameters such as intensity, energy transfer, lifetime, and quantum efficiency.
A highly ultrasensitive fluorescence sensor should be developed for detecting the appropriate fluorescence molecules, laser power regulation, and virus detection” I don’t see what you mean with laser power regulation.
The change in the fluorescent signal of the probe is measured to detect the virus. However, photolulling and photobleaching usually occur with fluorescent molecules under high-power laser irradiation or long-term illumination. Therefore, an ultrasensitive fluorescent sensor must be constructed using suitable fluorescent molecules with adjustable laser power.
“The optical properties of graphene and GO are of interest and somewhat unknown.” Please be more specific.
The graphene and GO samples obtained so far are generally micron-sized or larger. Finally, little is known experimentally about the properties of graphene with molecular dimensions of about 10 nm or less.
“their importance has reduced due to some disadvantages” please be specific.
Although spherical carbon nanomaterials have attracted attention in medicine due to their immunogenicity, biocompatibility, uniform surface chemistry, and simple geometry, their importance has decreased due to some disadvantages such as poor water solubility, lack of biodegradability, toxicity concerns, and low pK.
- Enumeration starting line 269 should be reworked, or turned into a clear sentence. A few words should be sufficient to say the QDs should be affordable, enable highly sensitive assays and not display toxicity.
However, certain global properties such as being easy to measure, providing rapid results, and being non-invasive and cost-effective are essential for all QD-based biomarkers.
- “QD-based biomarkers can be categorized into four groups on the basis of their contribution to business, regulatory, and clinical decision-making” I really don’t see where the authors want to go with that opening sentence. At the end of the paragraph, four categories are mentioned again: “Figure 8 shows the classification of QD-based biomarkers into four categories: target, mechanism, pathophysiological, and detection. Could these categories be clarified or corrected if needed? The flow chart and explanation is likely to match general/textbook medical/pharmaceutical knowledge of the positioning of drugs and/or diagnostics in patient care, with not much specificity related to QDs, so this part and figure could be removed.
This paragraph was removed.
- “To target the infamous spike protein on the coronavirus,” infamous could be removed remain objective.
Done.
- “researchers designed experiments in which graphene sheets, which are more than 1,000 times thinner than postage stamps,” The authors should provide precise measurements and units.
The researchers have developed experiments combining graphene sheets that are more than 1,000 times thinner than postage stamps (thickness of 0.34 nm) with antibodies.
7 “QDs are paired with many biological molecules such as proteins, drugs, DNA probes, and oligonucleotide probes due to the modifiable nature of semiconductor nanocrystals with regard to their surface.”, “The ability of QDs to quickly change their level as well as their faster and cheaper detection mode make them a successful tool for any diagnosis” similar statements are repeated many times in the article. Once or twice in the text is sufficient, then focus should be made on specific information. The authors should try to avoid too many repetitions of basic statements, in particular when describing analytical approaches to detect different pathogens.
Response:
- All the pathogens cited are well known and extensively reviewed, and the focus of this review is on QDs, so the pathogens/medical conditions should be more briefly described, and the readers referred to specialised reviews.
5.1. Coronavirus disease‐2019
Coronavirus disease-2019 (COVID -19) causes severe complications and deaths in humans. To combat COVID -19, extensive studies are being conducted to detect the virus in time to reduce the alarming mortality rate associated with the infection. QDs are multifunctional, small nanoparticles that can effectively act as biosensors and potential targets for cancer cells and viruses. QDs, also known as "semiconductor nanomaterials", can be combined with highly fluorescent probes, which are important for long-term fluorescence diagnosis and imaging of various cellular processes [168, 169]. The size of QDs is adjustable from 1 to 10 nm with an optical wavelength comparable to adjustable plasmonic nanoparticles (10 to 300 nm). Therefore, QDs have been identified as a new fluorescent probe for molecular imaging [170, 171]. Due to these exceptional properties, QDs can be considered an important factor in diagnosing viruses and combating viral infections. In addition, the integration of potential biocompatible carriers may facilitate interdisciplinary research and enable clinical approaches to combat the virus. The elimination and diagnosis of SARS-CoV-2 infections by QDs has attracted great interest from researchers. The initial reasons for using QDs can be attributed to their ability to be tracked under a specific wavelength of light [172, 173]. Moreover, QDs can effectively targetand penetrate SARS-CoV-2 with a size between 60 and 140 nm [174, 175]. Separation/inactivation of the S protein from SARS-CoV-2 is possible due to the positive surface charge of carbon-based QDs [176]. Furthermore, the production of reactive oxygen species within SARS-CoV-2 occurs through the interaction of the cation surface charges displayed by QDs with the negative RNA strand of the virus [177, 178]. The antiviral effect of carbon dots (CD) against porcine reproductive and respiratory syndrome virus and pseu-dorabies virus was studied by Du et al [179]. Activation of interferon-stimulated genes, especially interferon-α production, is induced by CDs that suppress viral replication. Furthermore, integration of functional targets with QDs can effectively interact with SARS-CoV-2 entry receptors and influence genomic replication [124].
- Boles, J., et al., Relationship selling behaviors: antecedents and relationship with performance. Journal of business & industrial marketing, 2000.
- Mudshinge, S.R., et al., Nanoparticles: emerging carriers for drug delivery. Saudi pharmaceutical journal, 2011. 19(3): p. 129-141.
- Peer, D., et al., Nanocarriers as an emerging platform for cancer therapy. Nano-Enabled Medical Applications, 2020: p. 61-91.
- Hashemi, S.A., et al., Ultrasensitive Biomolecule‐Less Nanosensor Based on β‐Cyclodextrin/Quinoline Decorated Graphene Oxide toward Prompt and Differentiable Detection of Corona and Influenza Viruses. Advanced Materials Technologies, 2021. 6(11): p. 2100341.
- Jha, S., et al., Pharmaceutical potential of quantum dots. Artificial cells, nanomedicine, and biotechnology, 2018. 46(sup1): p. 57-65.
- Omidifar, N., et al., Different Laboratory Diagnosis methods of COVID-19: A Systematic Review. Archives of Clinical Infectious Diseases, 2021. 16(1).
- Prajapat, M., et al., Drug targets for corona virus: A systematic review. Indian journal of pharmacology, 2020. 52(1): p. 56.
- Omidifar, N., et al., Different Laboratory Diagnosis Methods of COVID-19: A Systematic. 2021.
- Ting, D., et al., Multisite inhibitors for enteric coronavirus: antiviral cationic carbon dots based on curcumin. ACS Appl Nano Mater 1: 5451–5459. 2018.
- Dong, X., et al., Carbon dots’ antiviral functions against noroviruses. Scientific reports, 2017. 7(1): p. 1-10.
- Chen, L. and J. Liang, An overview of functional nanoparticles as novel emerging antiviral therapeutic agents. Materials Science and Engineering: C, 2020. 112: p. 110924.
- Du, T., et al., Carbon dots as inhibitors of virus by activation of type I interferon response. Carbon, 2016. 110: p. 278-285.
- Iannazzo, D., et al., Graphene quantum dots based systems as HIV inhibitors. Bioconjugate chemistry, 2018. 29(9): p. 3084-3093.
5.2. HIV
The presence of boronic acid on QDs as a targeting ligand can be specifically activated in the viral cycle and benefits from the better interaction of QDs with covalent, non-covalent and electrostatic interactions, as well as their optical properties, which enable their use in HIV diagnosis [198]. Iannazzo et al. investigated the role of modifying QD-based nanomaterials in increasing their potency against HIV-1. The results also showed that modifying graphene quantum dots or QDs with non-nucleoside reverse transcriptase inhibitors (NNRTIs) could increase their activity in diagnosing HIV-1 [124]. To detect HIV dsDNA in serum, a relative fluorescence biosensor consisting of CQD and cadmium telluride quantum dots (CdTe QDs) was developed by Liang et al. CdTe QDs coated with mercapropionic acid were first coupled with mitoxantrone (MTX), a synthetic anthraquinone drug that can interfere with DNA, resulting in red fluorescence quenching at 599 nm due to electron transfer between CdTe QDs and MTX. In such a situation, only fluorescence emission at 435 nm can be detected due to the green QDs. Eventually, the CdTe QDs have moved further away from the MTX -ssDNA complex due to electrical repulsion. Also in the presence of HIV-dsDNA, the specific binding of MTX to dsDNA resulted in the separation between MTX and CdTe QDs [199].
- Szunerits, S., et al., Nanostructures for the inhibition of viral infections. Molecules, 2015. 20(8): p. 14051-14081.
- Iannazzo, D., et al., Graphene quantum dots based systems as HIV inhibitors. Bioconjugate chemistry, 2018. 29(9): p. 3084-3093.
- Liang, S.-S., et al., Ratiometric fluorescence biosensor based on CdTe quantum and carbon dots for double strand DNA detection. Sensors and Actuators B: Chemical, 2017. 244: p. 585-590.
5.3. HPV
- Xue et al. used quantum dot (QD) in situ hybridization (ISH) in studying the association of oral squamous cell carcinoma with HPV. After comparing QDISH with conventional ISH, the presence of high-risk HPV16/18 was detected. Thus, the results suggest that QD may be an effective method for detecting HPV infection and HPV-related oral squamous cell carcinoma [215]. Piao et al. also investigated different types of fluorescent nanomaterials such as metal nanoparticles, QDs and fluorophorodised silica nanoparticles to detect human papillomavirus DNA [216]. Nejdl et al. analysed the interactions between blue and yellow fluorescent CdS QDs to decipher the HPV-16 oncogene E6. The results showed that yellow-low fluorescent CdS-QDs showed a higher affinity for DNA (E6 HPV-16) than blue dyes [217].
- Xue, J., et al., Use of quantum dots to detect human papillomavirus in oral squamous cell carcinoma. Journal of oral pathology & medicine, 2009. 38(8): p. 668-671.
- Piao, J.Y., et al., Direct visual detection of DNA based on the light scattering of silica nanoparticles on a human papillomavirus DNA chip. Talanta, 2009. 80(2): p. 967-973.
- Nejdl, L., et al., Interaction of E6 gene from human papilloma virus 16 (HPV-16) with CdS quantum dots. Chromatographia, 2014. 77(21): p. 1433-1439.
5.4. Hepatitis
Chowdhury et al. investigated an electrochemical sensor based on enhanced QDs for the detection of hepatitis E virus (HEV). The results showed that the proposed sensor based on QDs could pave the way for the development of robust and powerful assay methods for the detection of HEV [233]. Liu et al. investigated a protein array based on QDs encoded microbes for the diagnosis of hepatitis C virus. 120 HCV-positive and 50 HCV-negative samples were tested to evaluate the protein array based on QDs encoded microbes and compare it with the results of the recombinant immunoblot assay (RIBA) as the gold standard [234]. Wang et al. investigated the highly efficient production of QD barcodes at multiple scales for the diagnosis of multiplex hepatitis B. Five HBV indicators were selected as diagnostic targets to investigate the feasibility of QD barcodes in high-performance detection. The results showed that these QD barcodes are very useful for the simultaneous and selective detection of many different biomolecular targets [235].
- Chowdhury, A.D., et al., Electrical pulse-induced electrochemical biosensor for hepatitis E virus detection. Nature communications, 2019. 10(1): p. 1-12.
- Liu, J. and G.-X. Zhang, A protein array based on quantum dots (QDs) encoded microbeads for detection of hepatitis C virus. Zhonghua shi yan he lin Chuang Bing du xue za zhi= Zhonghua Shiyan he Linchuang Bingduxue Zazhi= Chinese Journal of Experimental and Clinical Virology, 2013. 27(1): p. 67-69.
- Wang, G., et al., Highly efficient preparation of multiscaled quantum dot barcodes for multiplexed hepatitis B detection. Acs Nano, 2013. 7(1): p. 471-481.
- The conclusion and perspective gives mostly very generic statements with low added value. This should be reworked to extract the most interesting, specific information and take home messages from the assembled literature, to give the readers the feeling to have learned something they do not already know, or a perspective they did not yet experience.
Thanks for the good comments about this section. We have re-examined the article carefully. We have re-write the conclusion and perspective.
Early detection of viral infections and their effective treatment is key to reducing viral infections and improving treatment success rates and best chances. It is well known that molecular-based tests are not very sensitive, require a long time to perform and do not allow on-site detection of viral infections in biological media. Currently, the majority of routine viral diagnoses are based on molecular tests. Quantum dot biomarkers, which enable miniaturised biosensors for effective early diagnosis of viral infections in the field, have shown excellent potential for revolutionising viral infection. Strong covalent binding between these molecules has led to advanced sensor substrates that can be used for precise detection of viral biomarkers and monitoring of treatment progress at the point of care.In this review, we describe different approaches for the synthesis and functionalisation of QDs by incorporating various biomolecules of interest that are able to recognise and convert them into signals, such as antigens, enzymes, hormones, proteins and virus-associated by-products, as well as biomolecules on the surface of viruses. Multiplex biosensors have been developed to study the effectiveness of virus detection systems. The results of these tests have shown that these tools are useful in diagnosing infections as well as evaluating the efficacy of viral surveillance, indicating their potential use in the clinical setting to detect and treat viral infections. Although the multiplex biosensors developed have excellent performance, future research in this area must take into account that integrated biosensors can be used as part of point-of-care systems to detect multiple biomarkers simultaneously. In addition, laboratory instruments should be able to provide accuracy and reliability while being sensitive, fast and convenient to use. The development of QD biomarkers or biosensors is a major challenge due to the synthesis of high quality and stable QDs with defined size, shape, charge and degree of agglomeration. These properties have a great impact on the physicochemical properties and performance of the QD biomarkers. In order to compare the results obtained with those of other existing techniques, further studies are needed to explore the outstanding results obtained with these nanomaterials for biosensing. A new protocol for the detection of viral biomarkers with these highly sensitive biosensors needs to be developed. With the right combination of knowledge from the fields of biology, physics, chemistry and engineering, advanced QD-based biomarkers will enable simple, rapid and accurate early and multiplex diagnosis of various viral infections in the clinic in the near future.
Round 2
Reviewer 2 Report
The text style and organization have been much improved, congratulations. Some remaining comments:
Some good work has been done in the way of graphical improvement, although some elements such as QDs are still stretched out of proportions in many figures.
Lines 208-211 - one sentence is repeated.
Line 221 - Photolulling: Is this a new term, could you explain?
Line 333 - there is a jump from CNTs to CQDs, maybe you need to add a transition from one to the other, or an additional part 3.5 on CQDs.
Line 430 - 0.0048 μAμg.mL−1. cm−2 , pleaser check the unit.
Author Response
Dear reviewer,
Pharmaceuticals journal
Thank you for considering our manuscript. We would like to appreciate for appending the reviewers' comments. We have carefully reviewed the comments and have revised the manuscript, accordingly. Our responses are given in a point-by-point manner below. Changes in the manuscript has been highlighted in yellow font color.
We hope the revised version is now suitable for publication and look forward to hearing from you in due course.
Sincerely,
Dr. Ahmad Gholami
on behalf of all authors
Reviewer 2
The text style and organization have been much improved, congratulations. Some remaining comments:
Some good work has been done in the way of graphical improvement, although some elements such as QDs are still stretched out of proportions in many figures.
Response: Thanks for your valuable suggestion, it has been corrected.
Lines 208-211 - one sentence is repeated.
Response: Thanks for your valuable suggestion. The repeated part was removed.
Line 221 - Photolulling: Is this a new term, could you explain?
Response: Thank you for your consideration and your great comment, photo quenching is the correct word that was placed.
Line 333 - there is a jump from CNTs to CQDs, maybe you need to add a transition from one to the other, or an additional part 3.5 on CQDs.
Response: Thanks for your valuable suggestion. This is really a high-level good comment, corrected.
Since CNTs have low solubility in water and have difficulty in providing strong fluorescence in the visible region, this greatly limits their application. Therefore, CQDs as new nanomaterials based on zero-dimensional carbon, which are known for their small size and relatively strong fluorescence properties, have been considered in various applications such as biomedicine [97].
- Wang, X., et al., A mini review on carbon quantum dots: preparation, properties, and electrocatalytic application. Frontiers in Chemistry, 2019. 7: p. 671.
Line 430 - 0.0048 μAμg.mL−1. cm−2 , pleaser check the unit.
Response: Thanks for your valuable suggestion, was checked and 0.0048 μAμg.mL−1.cm−2 is correct.